# EV-Eye: Rethinking High-frequency Eye Tracking through the Lenses of Event Cameras

**Guangrong Zhao[1], Yurun Yang[1], Jingwei Liu[1], Ning Chen[1],**
**Yiran Shen[1]\*, Hongkai Wen[2] and Guohao Lan[3]**
[1]School of Software, Shandong University, China
[2]Department of Computer Science, University of Warwick, UK
[3]Department of Software Technology, Delft University of Technology, NL
{guangrong.zhao, yiran.shen}@sdu.edu.cn
hongkai.wen@warwick.ac.uk
g.lan@tudelft.nl

## Abstract

In this paper, we present **EV-Eye**, a first-of-its-kind large-scale multimodal eye tracking dataset aimed at inspiring research on high-frequency eye/gaze tracking. EV-Eye utilizes the emerging bio-inspired event camera to capture independent pixel-level intensity changes induced by eye movements, achieving submicrosecond latency. Our dataset was curated over two weeks and collected from 48 participants encompassing diverse genders and age groups. It comprises over **1.5 million** near-eye grayscale images and **2.7 billion** event samples generated by two DAVIS346 event cameras. Additionally, the dataset contains **675 thousand** scene images and **2.7 million** gaze references captured by a Tobii Pro Glasses 3 eye tracker for cross-modality validation. Compared with existing event-based high-frequency eye tracking datasets, our dataset is significantly larger in size, and the gaze references involve more natural and diverse eye movement patterns, i.e., fixation, saccade, and smooth pursuit. Alongside the event data, we also present a hybrid eye tracking method as a benchmark, which leverages both the near-eye grayscale images and event data for robust and high-frequency eye tracking. We show that our method achieves higher accuracy for both pupil and gaze estimation tasks compared to the existing solution.

## 1 Introduction

Eye tracking is a technique that continuously measures the movement of the eyes [1] and has shown great promise in a wide range of scientific fields [2] and everyday applications [3]. Traditional eye tracking systems in current mainstream leverage traditional CCD/CMOS cameras to capture the appearance of eyes for tracking [4, 5, 6, 7, 8]. Unfortunately, constrained by the frame rate and the limited bandwidth of CCD/CMOS camera, the tracking frequency of conventional eye tracking systems is often bounded by a few hundred hertz, e.g., the latest Tobii Pro Glasses 3 eye tracker [9] has a tracking frequency of 100Hz. Similarly, Pupil Labs [10] achieves up to 200Hz gaze estimation. Although such a modest frequency is enough for many common use cases, e.g., human-computer interaction [11, 12] and activity recognition [13, 14], it stands in the way of enabling *game-changing* applications that may require over kilohertz tracking frequency, such as the diagnosis of mental disorders [2, 15], and gaze-based user authentication [16, 17]. The peak angular speed of the human eye in saccade can reach up to $700°/s$ [18] and the eye's acceleration is up to $24,000°/s^2$ [19]. The high-frequency eye tracking can reveal latent information encoded in high-speed and irregular

---

\*Corresponding Author

37th Conference on Neural Information Processing Systems (NeurIPS 2023) Track on Datasets and Benchmarks.

eye movements. For example, during the diagnosis of mental disorders, an increase in delays for visually guided saccades can help to characterize ADHD [20]. Unfortunately, achieving accurate eye tracking beyond kilohertz frequency requires a significant increase in camera bandwidth, which becomes a fundamental barrier for mainstream CCD/CMOS camera-based systems. Some high-end eye tracking systems (costing tens of thousands of US dollars), such as the EyeLink 1000 [21], can provide eye tracking at 1KHz by utilizing high-speed cameras. However, the high frame rate presents a considerable computational burden on downstream processing.

This challenge has inspired the use of the emerging bio-inspired dynamic vision sensor [22, 23], also known as the event camera, for eye tracking [24, 25]. In contrast to traditional CCD/CMOS cameras, which acquire information in a frame-based principle with a fixed frame rate, an event camera perceives the scene by capturing independent pixel-level light-intensity changes and producing asynchronous event streams that indicate the locations and polarity of the intensity changes. Due to its asynchronous nature and cost-saving readout, an event camera can achieve sub-microsecond latency [22]. Moreover, an event camera operates adaptively: the faster the targeted motion, the more events are generated per second, and vice versa. In a near-eye scenario, the light-intensity changes induced by eye movement are sparse in both time and space [25]. Thus, the event camera can adapt to the density of events based on the speed of eye movements and utilizes the camera bandwidth more efficiently than traditional cameras. These properties make event cameras ideal for high-frequency eye tracking.

While event-based eye tracking has demonstrated significant advantages over its frame-based counterpart, it is still in its early stage and requires substantial research efforts and public resources, such as datasets. First, similar to traditional camera-based eye tracking [26, 8, 27, 28], the success of event-based eye tracking heavily relies on the availability of large-scale event-based datasets. However, as detailed later in Section 3, the only existing dataset, EBVEYE [25], provides sparse gaze references with a limited range of eye movements (mainly fixation) collected from a limited number of subjects. This limitation poses a risk of high sensitivity to subject and tracking condition changes. Secondly, the state-of-the-art event-based eye tracking solution adopts the model-based approach [25], which is not robust to subject diversity and sensor noise. This work aims to shed light on the existing challenges and advances the field of event-based eye tracking by making the following contributions:

• We introduce the largest and most diverse multi-modal frame-event dataset for high-frequency eye tracking in the literature (over 170Gb in total).[2] To the best of our knowledge, ours is the largest multi-modal dataset aimed at high-frequency eye tracking in the literature.
• We propose a novel hybrid frame-event eye-tracking benchmarking approach tailored to the collected dataset, capable of tracking the pupil at a frequency up to 38.4KHz. As demonstrated using our dataset, the proposed approach significantly outperforms the existing solution [25] in both pupil and gaze estimation by a large margin.

## 2   Related Work

**Applications of Event Camera.** The methods of event-based applications normally start by converting the asynchronous event streams into formatted representations. For example, image-like representations convert event streams into event-based "images" by calculating the distributions of the events and their timestamps. Image-like representations have been widely used for object tracking [29], gait recognition [30], and optical flow estimation [31]. To better preserve the temporal information of event streams, graph-based representations [32, 33], point cloud-based representations [34] and voxel-grid-based representations [35] have been proposed. Convolutional neural networks, graph-based convolution, and PointNet [36] can be applied to the corresponding representations for various vision tasks, including object/ action classification [32], gait recognition [33], gesture recognition [34], frames interpolation [37] etc.

**Conventional Eye Tracking.** Current solutions for eye-tracking are either model-based [38, 39] or appearance-based [5, 26]. The model-based methods rely on corneal reflection or pupil shape to infer the Point of Gaze (PoG) from parametric features, such as pupil centers and iris. However,

---

[2]The EV-Eye dataset and the implementation of our benchmarking methodologies can be found at: https://github.com/Ningreka/EV-Eye.

these methods are sensitive to ambient light conditions and require high-resolution images to ensure good tracking accuracy. By contrast, appearance-based methods leverage neural networks to map images of eyes to the PoG, which have shown significant improvement in gaze estimation accuracy and system robustness [5, 26]. However, limited by the frame rate of conventional cameras, the eye tracking frequency of existing methods is usually constrained to around 200Hz. In addition, existing conventional eye-tracking datasets rarely contain videos, and when they do, they are only of participants gazing at fixed points [40]. While there are some datasets including natural eye movement patterns such as saccades, fixations, and smooth pursuits [41, 42, 43], they do not provide data in high temporal resolution and dense gaze references.

**Event-based Eye Tracking.** There are also works that leverage event streams for eye tracking. For instance, Feng et al. [44] propose an event-driven eye segmentation method for eye tracking. However, the event streams were generated by the event simulator rather than collected in real-world experiments. Ryan et al. [45] propose an event-based neural network for real-time face and eye detection. However, their focus is primarily on the detection and localization of eyes, rather than pupil tracking. More recently, Stoffregen et al. [24] introduce a coded differential lighting method to track eye movement at a frequency of 1KHz. Angelopoulos et al. [25] is the most relevant work to ours, in which the authors propose a hybrid frame-event approach that achieves up to 10KHz eye tracking. However, our method exhibits greater robustness across user diversity and achieves higher accuracy in eye tracking. Additionally, our dataset consists of 48 subjects, which is twice that of EBVEYE. Lastly, our dataset includes dense gaze references for multiple eye movements i.e., fixation, saccade, and smooth pursuit (the definition can be found in Section 3.2 Eye Tracking Model), while EBVEYE only provides sparse references during fixation status.

## 3 The EV-Eye Dataset

### 3.1 Dynamic Vision Sensors

To make the paper self-contained, we will brief the principal background of event cameras. Unlike traditional RGB cameras, event cameras do not produce synchronous video frames at a fixed rate, but asynchronous event streams. Specifically, pixels of the event camera work independently, to detect the change in the light intensity of the scene as,

$$|log\ I(x, y, t_{now}) - log\ I(x, y, t_{previous})| < C \qquad (1)$$

where $I(x, y, t)$ is the intensity value of pixel $(x, y)$ at time $t$. When the change of intensity at the pixel is over the threshold $C$, an event will be released immediately. An event stream is a collection of events over time and is represented as a stream of quadruplet $\{x, y, t, p\}$. When the event corresponds to a positive change, the polarity $p$ is $+1$ otherwise it is $-1$. Compared with traditional RGB cameras, event cameras possess several unique characteristics. As an event is launched as soon as a change is detected without global synchronization, the event streams are high in temporal resolution and low in response latency (in the order of microseconds). Event cameras save sensing energy and bandwidth as they produce events only when changes are detected. The high dynamic range (140 dB vs. 60 dB of traditional RGB cameras) enables them to work greatly under challenging lighting conditions. These characteristics make event cameras have great potential for high-speed motion capture and working on resource-constrained devices.

### 3.2 Preliminaries

**Eye Tracking Model.** In this paper, we consider a common eye-tracking model. It starts by localizing the centroids of the pupil area in the image domain. Subsequently, the corresponding Point of Gazes (PoGs) are determined, representing where the users are looking in realworld, through polynomial regression [46]. Eye movements are various and can be vastly categorized into three major types in natural viewing scenarios [41], i.e., fixation, saccade, and smooth pursuit. **fixation** is the status of eyes when users stare at a fixed point; **saccade** represents the fast eye movement towards a point of interest; **smooth pursuit** is the eye movement when the users follow a smoothly moving object in a predictable route.

**Eye Tracking Modalities.** As shown in Figure 1(a), EV-Eye adopts three different sensing modalities. These modalities include near-eye grayscale images and event streams captured by two sets of DAVIS346 event cameras [47], and gaze references provided by Tobii Pro Glasses 3 [9].

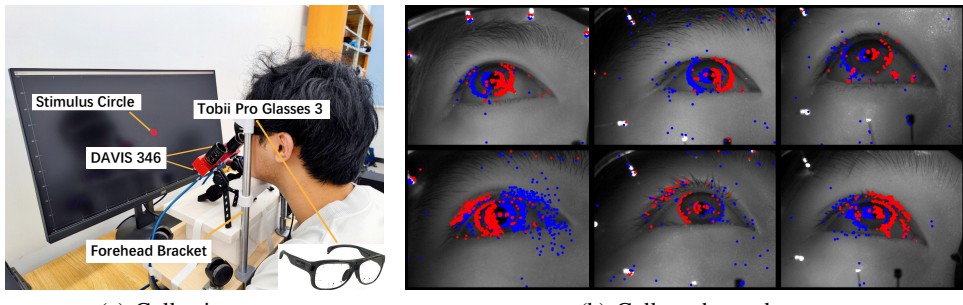

| (a) Collection setup | (b) Collected samples |

Figure 1: **Illustration of dataset collection setup (a) and collected samples (b)**. The near-eye grayscale images overlaid with 40ms of events.

*Event Streams:* The event streams are collected by two sets of DAVIS346 event cameras with a resolution of 346×240. They comprise positive and negative events triggered by intensity changes resulting from eye movements, winks, and other subtle motions. This setup provides high temporal resolution, enabling high-frequency eye tracking.

*Grayscale Images:* The DAVIS346 cameras also record near-eye grayscale image sequences at a frame rate of 25fps. They complement the event streams by providing rich semantic information about the eyes and facilitating robust pupil segmentation in the image domain.

*Gaze References:* As obtaining ground truth for Points of Gaze (PoGs) during eye movement is not feasible, we adopt a commercialized device, Tobii Pro Glasses 3 [9], to acquire gaze references. It provides the PoGs and pupil diameters of the users at 100Hz. Tobii Glasses have a field of view (FoV) of 95°×63°, and it can achieve 0.6 angular error in gaze estimation in the central area of the virtual screen.

### 3.3 Dataset Curation

**Data Collection Setup.** The setup of the data collection is illustrated in Figure 1(a). During the collection, the subject's head is securely fixed on an ophthalmic headrest to prevent any unintended movement. Two DAVIS346 event cameras are positioned closely to record the movement of each eye. The aperture and exposure time of DAVIS346 are adjusted to prevent overexposure. We leverage a 32-inch monitor with a resolution of 1920×1080 to display the visual stimulus to guide the gaze movement of the subject. The distance between the monitor and the subject is about 33cm, which leads to a field of view (FoV) of 95°×63°. The visual stimulus is a solid red circle displayed on the monitor, with a diameter of 60px. In our setup, the subject wears Tobii Glasses to obtain reference PoGs. Additionally, the scene camera of Tobii records scene images throughout the experiments. All devices are synchronized before each data collection session.

**Acquisition Protocol.** We recruited 48 participants (28 male and 20 female)[3] aged between 21 and 35 years. They have various vision corrections, and each of them participates in four sessions of data collection. For the first two sessions, the screen is equally divided into $11 \times 11$ grid, resulting in 121 squared blocks. The stimulus appears in the center of one block in a random sequence with each appearance lasting for 1.5s. The subject quickly finds the stimulus and focuses on it when it is stationary. The random sequence for each subject is the same, and each session ends when all blocks are visited. These two sessions allow us to trigger and capture both saccade and fixation states of the eye movement. For the last two sessions, we ask the subject to fixate on a smoothly moving stimulus across the screen. The stimulus first moves horizontally from the lower-right to the upper-left corner of the screen then it moves vertically from the upper-left to the lower-right. The movement of the stimulus is a predictable squared wave trajectory. The space between the adjacent horizontal trajectories is 54px, which results in 20 horizontal paths. Similarly, the space between adjacent vertical trajectories is 96px and leads to 20 vertical paths. The moving pattern of the stimulus allows us to record eye movement in smooth pursuit.

---

[3]Our data collection was approved by our local IRB council.

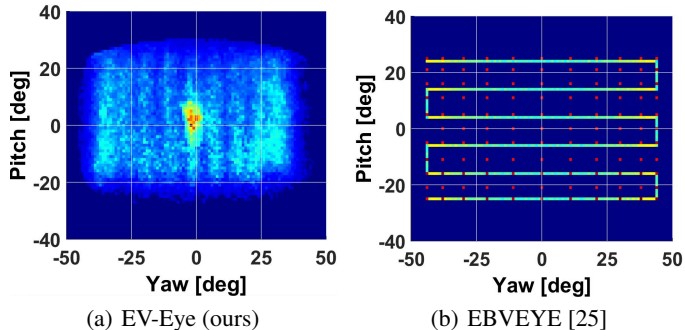

| (a) EV-Eye (ours) | (b) EBVEYE [25] |

Figure 2: **Distributions of the gaze references of datasets, the gaze references provided in our dataset are significantly more dense and involve all states.**

**Data Annotation.** We leverage the VGG Image Annotator [48] to label the pupil region of 9,011 near-eye images selected uniformly across the whole image dataset. Normally, the pupil region is regarded as an ellipse. Therefore, we label the region by adjusting the major axis, minor axis, tilt angle, and center of the ellipse [49]. Then inpolygon function [50] is applied to generate binarized masks $G$ as the ground truth according to the region of the ellipse. Besides the data collected by DAVIS346, our dataset also includes 2.7 million PoGs estimated by the Tobii Pro eye tracker and 675 thousand images recorded by its scene camera. As the ground truth of gaze is impossible to obtain, data collected by Tobii is regarded as a reference for gaze estimation.

### 3.4 Data Characteristics

Our dataset includes multi-modal data collected from two DAVIS346 cameras and one Tobii Pro Glasses 3. The two DAVIS346 cameras produce 1.5 million near-eye grayscale images and more than 2.7 billion events. Figure 1(b) shows some samples of near-eye grayscale images from nine subjects. The images are overlaid with 40ms accumulated events between two consecutive frames.

Figure 2(a) demonstrates the distributions of the gaze references provided in our dataset. From the figure, we can observe PoGs of our dataset are densely distributed across a two-dimensional space with approx. 95° in yaw and 63° in pitch directions. However, as shown in Figure 2(b), EBVEYE [25] only provides very sparse gaze references. The red dots are the locations of the appearance of the stimulus for fixation state and the squared wave lines are the trajectory of the stimulus during smooth pursuit. EBVEYE assumes the human gaze can follow the guidance of the stimulus, though it is often not the case in practice. By comparing the two datasets, the gaze references provided in our dataset are significantly more dense and involve all states, i.e., fixation, saccade, and smooth pursuit. Then Tobii Glasses eye tracker can provide richer temporal information to enable researching on gaze estimation and eye movement dynamics [41]. It is worth noting that, in addition to the dynamic gaze references provided by Tobii, EV-Eye also provides the sparse coordinates of PoGs on the monitor screen during fixation states same as those "ground truth" provided by EBVEYE and static image-based datasets such as ETH-XGaze [51] and [52], the sparse coordinates are recorded by the scene camera of Tobii Pro Glasses 3 eye tracker synchronously[53].

## 4 Benchmarking Methodologies

In this section, we will describe our proposed benchmarking approach, which leverages both the near-eye grayscale images and the event streams generated by the event camera for accurate and high-frequency eye tracking. The overview of our method is shown in Figure 3.

### 4.1 Frame-based Pupil Segmentation

**DL-based Pupil Segmentation.** We employ U-Net [54] for pupil segmentation, which has been proven to achieve state-of-the-art accuracy and adopted as the backbone of several DL-based eye tracking works [55, 56, 57, 58]. The detailed settings of the model can be found in Appendix B. The pupil segmentation component outputs a binarized mask to extract the pupil area.

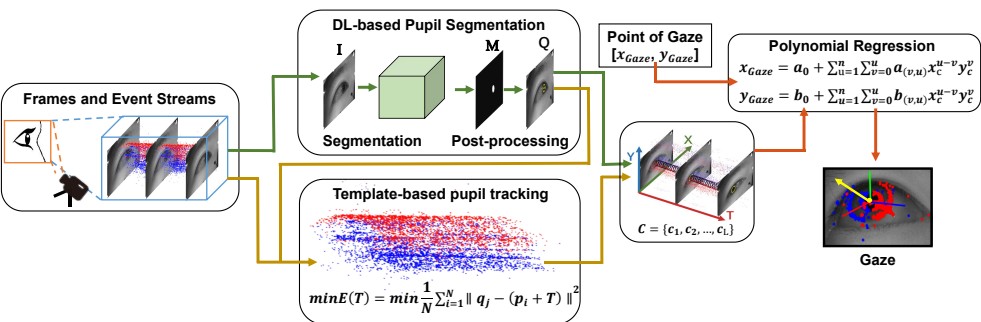

Figure 3: **Overview of the proposed benchmarking approach of EV-Eye**.

**Post-processing.** After obtaining the binarized mask $\mathcal{M}$, we adopt morphological closings [59] to remove additional noise, such as glint, in the segmented pupil area. Then, we consider the centroid of the segmented mask as the pupil center $c$ and employ an edge detector to find the pupil boundaries $Q$ as the pupil template for the following high-frequency tracking.

## 4.2 High-Frequency Event-based Pupil Tracking

**Candidate Points Subset.** After the pupil area is segmented out, we devise a method for selecting a subset of event points for high-frequency pupil tracking. This enables us to filter out noisy events caused by the movement of eyelashes and eyelids, which are unrelated to the actual pupil movement.

An example of the event points selection is shown in Figure 4(a). First, for each pixel on the boundary $Q$ of the pupil template, we calculate its distance $\gamma$ to the template center $c$. We denote the averaged distance for all pixels on $Q$ as $\bar{\gamma}$. Then, we select a set of $N$ event points to form a candidate point set $P$ with $|P| = N$ by the following rule:

$$\lambda_1\bar{\gamma} < ||(x, y) - c|| < \lambda_2\bar{\gamma}, \tag{2}$$

where $(x, y)$ is the coordinate of the current event point on the image; $||(x, y) - c||$ is the distance from the current event point to the template center; $0 < \lambda_1 < 1$ and $\lambda_2 > 1$ are two scale factors (we set $\lambda_1 = 0.8$, $\lambda_2 = 1.2$, respectively in our experiment). As shown in Figure 4(a), we accumulate $N$ event points lying between the two concentric circles to form the event points subset generated by pupil movement.

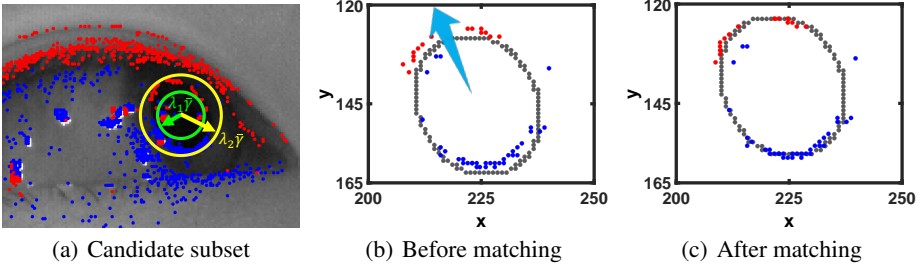

| (a) Candidate subset | (b) Before matching | (c) After matching |

Figure 4: **An example of selected candidate point subset (a)**: the event points lying between the two concentric circles form the candidate point subset and **an example of points-to-edge matching(b)**, the candidate point subset guides pupil updating, the blue arrow is the moving direction of the pupil.

**Points-to-edge Matching.** Next, we propose a points-to-edge matching approach to update the template center based on the accumulated candidate points set $P$.

We first calculate the translation of the current pupil boundary $Q$ using the candidate events set $P$. In our case, the events are generated due to the pupil movement in the horizontal and vertical directions in the camera space. As an example shown in Figure 4(b), when the pupil moves toward the upper-left corner, some of the iris pixels change to pupil pixels and generate red events with negative polarity (intensity decreases); similarly, some of the pupil pixels become iris pixels and produce blue events with positive polarity (intensity increases). Thus, the goal of the points-to-edge matching approach is

to find the optimal translation that optimizes the $\ell_2$-norm error $E$:

$$\min E(T) = \min \frac{1}{N} \sum_{i=1}^{N} \|q_i - (p_i + T)\|^2, \tag{3}$$

where $T$ is the optimal translation; $q_i$ and $p_i$ are samples in $Q$ and $P$, respectively. More specifically, $T$ can be obtained in three steps. First, for each event sample $p_i \in P$, we find its closest pixel $q'_{i*}$ in $Q$ by the nearest neighbor search and obtain $Q' = \{q'_1, ...q'_{i*}, ..., q'_N\}$, which is the set of nearest neighbors for all event points in $P$. Meanwhile, for each $p_i \in P$, we obtain its horizontal distance $\Delta T_{xi}$ and vertical distance $\Delta T_{y_i}$ to its nearest pixel $q'_{i*}$ in $Q$. In the second step, we shift each $p_i \in P$ along the horizontal and vertical axes by $\bar{\Delta T}_x$ and $\bar{\Delta T}_y$, respectively, where $\bar{\Delta T}_x = \frac{1}{N} \sum_{i=1}^{N} \Delta T_{xi}$ and $\bar{\Delta T}_y = \frac{1}{N} \sum_{i=1}^{N} \Delta T_{y_i}$. Finally, we repeat the first two steps and update the translation $T$ by:

$$T = T + \bar{\Delta T}, \tag{4}$$

where $\bar{\Delta T} = \{\bar{\Delta T}_x, \bar{\Delta T}_y\}$. The iteration terminates when $\bar{\Delta T}$ diminishes, i.e., $\bar{\Delta T}/T < 0.01$.

**Template Center Updating.** After obtaining the optimal translation $T$, we use the $N$ candidate events in $P$ to update the template center by:

$$c^{t+1} = c^t - T, \tag{5}$$

where $c^t$ is the last template center. By updating the center of the pupil template with a few events, the eye movement can be tracked in high frequency.

### 4.3  Gaze Estimation

After the pupil centers are obtained, we leverage the polynomial regression [46] to estimate the PoGs on the screen. Given the coordinate of the pupil center $c = (x_c, y_c)$, the corresponding PoG is obtained by $n$ order polynomial transformations:

$$x_{Gaze} = a_0 + \sum_{u=1}^{n} \sum_{v=0}^{u} a_{(v,u)} x_c^{u-v} y_c^{v}, \tag{6}$$

where $x_{Gaze}$ is the estimated horizontal coordinate of the PoG on the screen; $n$ is the polynomial order; $a_{(v,u)}$ and $b_{(v,u)}$ are the coefficients. The vertical coordinate $y_{Gaze}$ can be obtained following the same protocol. Due to the subject heterogeneity, e.g., different kappa angles and cornea radius of human eyes, the coefficients are obtained through a subject-dependent calibration.

## 5  Evaluation on EV-Eye Dataset

In this section, we implement two benchmarking approaches for evaluating the EV-Eye dataset: our proposed method and the model-based method introduced in EVBEYE [25].

### 5.1  Implementation and Evaluation Metrics

We implement the benchmarking methods with pytroch 3.8. When training the DL-based segmentation network, Adam [60] optimizer and ReduceLROnPlateau [61] scheduler are adopted. The initial learning rate is set to $1e^{-3}$, then it decays by multiplying a scale factor of 0.1 when the dice coefficient is not improved after two consecutive epochs. The batch size is set to 8. Both training and testing are implemented on an NVIDIA GeForce RTX 3090 GPU with 24GB VRAM.

Four metrics are adopted for the dataset evaluation (the detailed description can be found in Appendix Section C.1 ): **Intersection over union (IoU)** is a widely used metric for pupil region segmentation [62], which corresponds to the overlap between the estimated and groudtruth pupil region. **Dice coefficient (F1 Score)** is another commonly used metric for eye segmentation tasks [55, 63], which measures the similarity between the estimated and groundtruth pupil region. **Pixel error (PE) of eye tracking [8, 64, 65]** is the localization accuracy of eye tracking demonstrated as Euclidean distance in pixels between the estimated and groundtruth pupil centers. **Difference of direction (DoD) in gaze tracking** is the difference between the estimated and reference gaze directions to show the performance of gaze tracking. This metric is similar to the "gaze estimation error" on existing eye tracking literature [41, 52, 51].

## 5.2 Evaluation on frame-based pupil segmentation

Both our method and EVBEYE [25] contain the frame-based pupil segmentation component. The difference is that we adopt DL-based pupil segmentation with post-processing (DL-based method) rather than model-based method. The 9,011 manually-segmented images are used for the user-independent evaluation: in each round, the manually labeled images from one subject are selected for testing and those from the rest 47 subjects are used for training. The IoU, F1 score, and PE of each subject for two different methods i.e., DL-based method and model-based method are reported.

**IoU and F1 score.** The IoUs and F1 scores of different subjects obtained by the two methods are presented in Figure 5(a) and Figure 5(b), respectively. The DL-based method achieves significantly higher IoU and FI score for all the subjects compared with the model-based method. The average of IoU value is 0.9187 and 0.8360 for DL-based method and model-based method, respectively; while the average F1 score of these two methods are 0.9560 and 0.9075, respectively. In average, the IoU and F1 score of our DL-based method are $8.27\%$ and $4.85\%$ higher than the model-based method in pupil segmentation task.

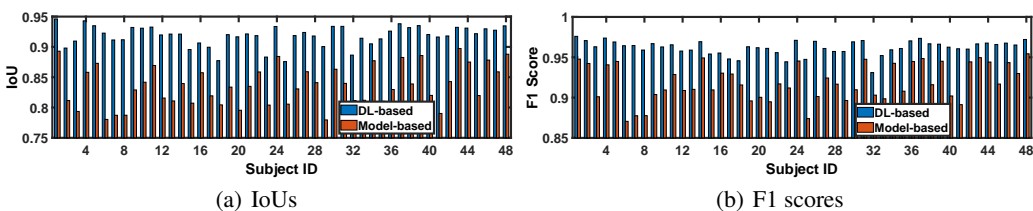

(a) IoUs  (b) F1 scores

Figure 5: **IoUs (a) and F1 scores (b) on frame-based pupil segmentation.**

**PE of frame-based pupil segmentation.** The PEs of the two methods are shown in Figure 6(a). Our DL-based method achieves significantly lower PE than model-based method for all subjects. The average PE of DL-based method and model-based method is 0.64px and 1.3px, respectively. Therefore, DL-based method achieves a significant 50.7% improvement in frame-based pupil segmentation over model-based method.

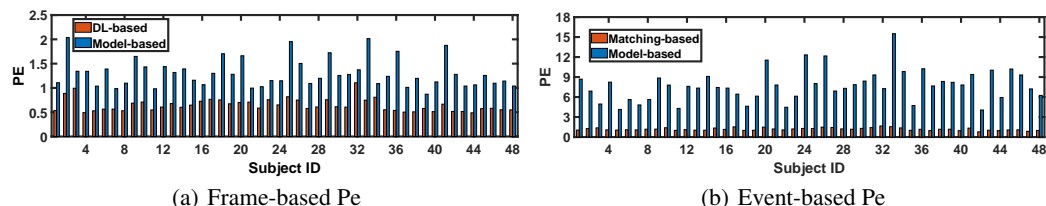

(a) Frame-based Pe  (b) Event-based Pe

Figure 6: **The pixel error of frame-based (a) and event-based (b) pupil tracking.**

## 5.3 Evaluation on event-based pupil tracking

Below, we compare and evaluate the accuracy of our template matching-based method and EVB-EYE [25]'s model-based method on event-based pupil tracking.

**PE of event-based pupil tracking.** As we are not able to obtain the event-wise groundtruth, we consider the 9,011 labeled frames as reference to assess the accuracy of event-based pupil tracking. Specifically, the two methods start with obtaining the pupil region of the last image before the labeled one. Then, event-based pupil tracking module runs through the events between the two images. The last pupil center obtained by the event-based module is compared with the groundtruth of the labeled image to obtain the tracking accuracy. The number of events for each event-based update is set to be 20 (same to that used in EVBEYE [25]). The PE of the two methods for each subject is shown in Figure 6(b). Our matching-based method achieves significantly lower PE for all subjects compared with model-based method. The average PE over all subjects is reduced by about 6.5×, i.e., from 7.7px to 1.2px.

## 5.4 Evaluation on update frequency

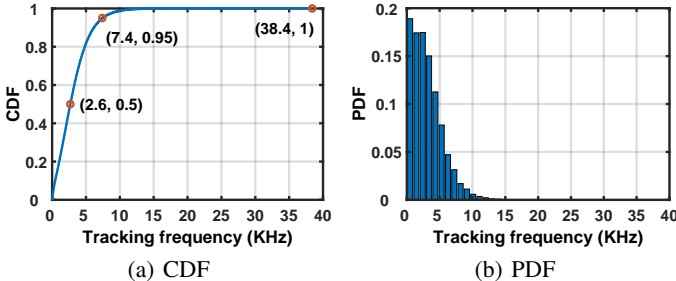

(a) CDF        (b) PDF

Figure 7: **CDF and PDF of instantaneous tracking frequency achieved by our method**.

We evaluate the temporal resolution of pupil tracking using EV-Eye by calculating the cumulative distribution function (CDF) and probability density function (PDF) of the tracking frequency. We update the pupil location as soon as a candidate points subset with 20 events is accumulated. The time difference between the first and last event is $T_{interval}$ and the instantaneous tracking frequency is defined as $F = 1/T_{interval}$. The CDF and PDF of tracking frequencies are presented in Figure 7. The tracking frequency of our method is dynamic because the time of generating 20 events are adaptive to pupil movement. Moreover, the peak tracking frequency $F_{peak}$ is up to 38.4KHz. This indicates our method is capable of capturing ultra-high-speed saccadic movement.

**Adaptive update strategy**: the tracking frequency of the benchmarking approaches is not at a fixed rate: it is determined by how long it takes to accumulate 20 events caused by the movement of the pupil region. Therefore, the tracking frequency is adaptive and proportional to the movement speed. 38.4KHz is the maximum tracking frequency calculated in our dataset (in saccade state). In fact, the extremely-high tracking frequency is an inherent property of an event camera which is high in temporal resolution (around tens of microseconds). Another benefit of the adaptive update strategy is its energy and computation efficiency: it is not necessary to work at very high frequency all the time. For example, when eyes are in a fixed or slow-moving state, the generated events are relatively few, leading to low updating frequency. This property can better utilize the limited resources on computation, energy, and bandwidth.

## 5.5 Evaluation on gaze tracking

The polynomial regression model (discussed in Section 4.3) is used to map the pupil centers in image domain to the PoGs for gaze tracking. As groundtruth is not available, we utilize the gaze references provided by Tobii for calibration following the similar protocol in [25]. We show the DoDs of the two methods for different subjects in Figure 8, since it's a general indicator on gaze estimation task [52, 41, 5, 66] . From the results we can observe, our method (Ours) achieves significantly lower DoDs for all subjects than model-based method from EVBEYE. The average DoD of Ours and model-based method for the whole field of view is $4.71°$ and $9.72°$ respectively, which indicates the gaze estimation results of our method are significantly closer to the commercialized eye tracker than model-based method proposed in EVBEYE.

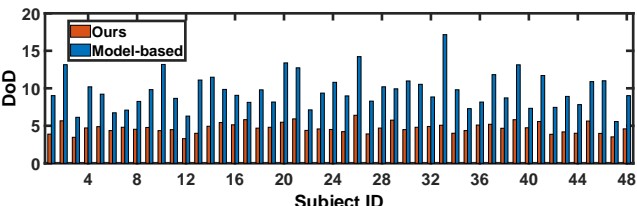

Figure 8: **DoDs of model-based method vs. ours with respect to the gaze references.** The average DoD of our method and model-based method for the whole field of view is $4.71°$ and $9.72°$ respectively.

# 6 Limitation

The collected multi-modal dataset, EV-Eye, and the proposed benchmark, as presented and evaluated above, offer accessible resources for research in developing new eye tracking algorithms with high frequency. These resources support applications that require the analysis of dynamic eye movement in high temporal resolution, such as the diagnosis of mental disorders. However, further efforts are required in the future to enhance the usability of the dataset. Firstly, additional labeling work is necessary to improve its usability. Currently, approximately 9,000 sparsely selected and labeled images are available to evaluate the accuracy of pupil region segmentation. To support a wider range of applications, a significantly larger number of images with diverse types of labels is required. For instance, we plan to label specific periods of different eye movement statuses, including saccade, fixation, smooth pursuit, blinks, etc. Furthermore, more texture details of the eye, such as regions of the pupil, iris, eyelids, and eyebrows, need to be labeled. Secondly, the dataset lacks ground truth for gaze. Obtaining ground truth for gaze tracking is challenging in real-world settings. Therefore, we utilized the commercial gaze tracker, Tobii Glasses3, to provide gaze references. However, a number of studies[67, 68] have shown that eye-trackers generally are less accurate for large view angles and this seems to be a common issue in many other eye-tracking systems such as ETH-XGaze, MPIIGaze and Angelopoulos et al [51, 52, 25]. Hence, methods to acquire more trustworthy and accurate gaze references need further exploration. Thirdly, regarding subject diversity, we are aware that at the moment the dataset is collected solely from an academic institution, which may introduce slight bias in some sense (e.g. no very diverse ethnicities). We plan to make every effort to expand our dataset to include more diverse participants. Lastly, in the context of the current work, we only select three major types of gaze behavior in natural viewing scenarios, i.e., fixation, saccade and smooth pursuit. However, we realize that considering more diverse scenarios in the real-world could enrich the gaze behavior in the dataset and thus broaden community use. In the next stage, we plan to use video stimuli such as CRCNS, DIEM [41], and Gaze360 [69]in the wild environment to capture more diverse types of eye movements.

# 7 Conclusion

In this paper, we introduce the most diverse and largest event-based multimodal dataset EV-Eye for high-frequency eye tracking, collected from 48 subjects with different devices. Frames and events from two DAVIS346 can describe the eye movement in extremely high temporal resolution and a commercialized eye tracker can provide densely distributed gaze references for cross modality comparison. Then we propose a novel hybrid frame-event eye-tracking approach to uncover the potential of the multi-modal dataset to achieve a tracking frequency of up to 38.4KHz. The extensive evaluations on EV-Eye demonstrate our method achieves significantly higher accuracy and is more robust to the diverse dataset than the state-of-the-art hybrid frame-event eye tracking method.

## Acknowledgments and Disclosure of Funding

This work is supported by Shandong Provincial Natural Science Foundation, China, Grant No.2022HWYQ-040.

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
