# Supplementary Material

## A   Access to Dataset and Benchmark

The dataset and the source code of the implementation of benchmark methods in this paper can be found via the link: `https://1drv.ms/f/s!Ar4TcaawWPssqmu-0vJ45vYR3OHw`, which will be released to the public when the paper is accepted.

## B   Pupil Segmentation Model

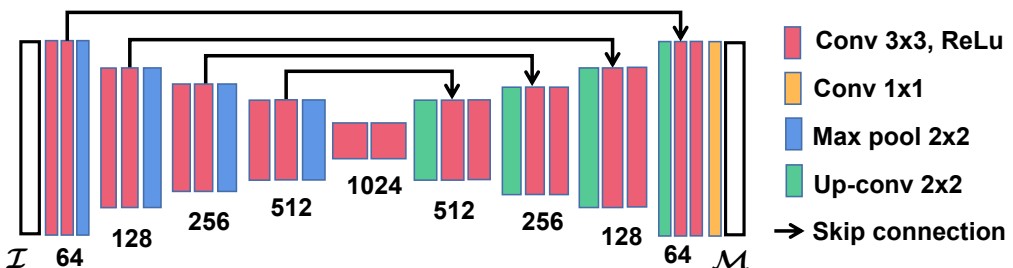

Figure 9: **Architecture of the U-Net based segmentation network.** The encoder takes an eye image $\mathcal{I}$ as the input for feature extraction, and decode generates the binarized feature mask $\mathcal{M}$.

To enable easy reproduction, we show the detailed architecture and parameter settings of the U-Net-based pupil segmentation model introduced in Section 4. As shown in Figure 9, it consists of an encoder and a symmetric decoder. The encoder is a stack of convolutional layers and max-pooling layers that extracts high-level semantic features from the input eye image $\mathcal{I}$. For each pair of convolutional layers, the convolution kernel size is 3 and the number of output channels is 64, 128, 256, and 512, respectively. The pooling size of the max-pooling layer is 2. The decoder aims to generate a binarized feature mask $\mathcal{M}$ for pupil segmentation. It has a symmetric structure with the encoder, but the max-pooling layers in the encoder are replaced by up-convolution layers. In addition, we use the skip connection in the segmentation network. This design allows the decoder to take the multi-scale and multi-level feature information extracted by the encoder into the segmentation process, and ensures a more refined segmentation result. We use the cross entropy loss and dice loss [71] for training. The loss function is given by:

$$\mathcal{L} = \frac{1}{J} \sum_{m=1}^{J} (-\sum_{k=1}^{K} G^{(k)} \log \mathcal{M}^{(k)} + 1 - \frac{2|\mathcal{M} \cap G|}{|\mathcal{M}| + |G|}), \tag{7}$$

where $G$ is the ground truth mask, $\mathcal{M}$ is the generated mask, $J$ is number of training samples in a batch, and $K$ is the number of categories. For pupil segmentation, $K = 2$, as the background and the pupil area are considered as two categories. An example of pupil segmentation result is shown in Figure 10.

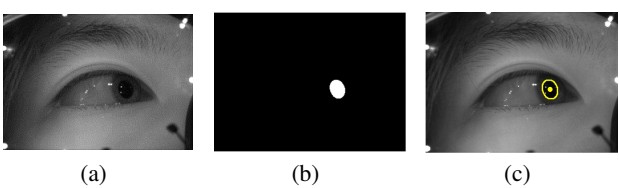

(a)  (b)  (c)

Figure 10: **Illustration of the frame-based pupil segmentation**: (a) the input eye image $\mathcal{I}$; (b) the generate binary mask $\mathcal{M}$; and (c) the detected pupil boundary $Q$ and the pupil center $c$.

# C More Details in Experiment

## C.1 Evaluation metrics

The detailed description of the four metrics adopted for the dataset evalution are as follows:

**Intersection over union (IoU).** IoU is widely used metric for pupil region segmentation [63], which is expressed as:

$$IoU(I, G) = (I \cap G)/(I \cup G), \tag{8}$$

where $I$ and $G$ are the estimated and groundtruth pupil regions respectively. The value of IoU ranges from 0 to 1, where 0 indicates the ground truth has no overlap with the estimated pupil region at all, whereas 1 indicates the groundtruth fully overlaps with estimated region.

**Dice coefficient (F1 Score).** F1 Score is another commonly used metric for eye segmentation task [56, 64], which measures the similarity between the estimated and groundtruth pupil region:

$$Dice(I, G) = (2|I \cap G|)/(|I| + |G|), \tag{9}$$

where $|I|$ and $|G|$ are the number of elements in the estimated region $I$ and the ground truth region $G$, respectively. The value of F1 Score ranges from 0 to 1, a higher value indicates better segmentation.

**Pixel error (PE) of eye tracking**. The localization accuracy of eye tracking is demonstrated as Euclidean distance in pixels between the estimated and groundtruth pupil centers. The pixel error is used for frame-based pupil segmentation and event-based pupil tracking in image domain.

**Difference of direction (DoD) in gaze tracking.** The difference between the estimated and reference gaze directions is adopted to show the performance of gaze tracking. Specifically, the distance in pixels ($P_d$) between the estimated and reference gazes is first calculated by projecting the gaze directions to the screen coordinates. Then DoD can be obtained by $DoD = \arctan(P_d/d_z)$ [42], where $d_z$ is the distance between the virtual screen to the scene camera of Tobii which can be easily obtained from the device.

## C.2 Further Discussion on Benchmarking Methodologies

In frame-based pupil segmentation stage, DL-based method shows superior accuracy for user-independent segmentation task than model-based method. The model-based method needs to be customized for each subject to produce reasonable results in our dataset, since it applies simple threshold-based approach [26] to distinguish the pupil area from its background. However, due to the diverse lighting conditions caused by different facial characteristics (nose bridge height, eye size, etc.), the threshold for each subject needs to be carefully tuned, which brings additional manual labor during the calibration stage. Moreover, background noise resulting from eye lashes, glasses frame, and eyebrows, significantly deteriorates the performance of the morphological opening used by model-based method. Therefore, one has to manually set a boundingbox for each subject to alleviate the influence of background noise. As shown in Figure 11, even if we adopt the personalized threshold and boundingbox, model-based method still fails under some challenging occasions, e.g., low contrast.

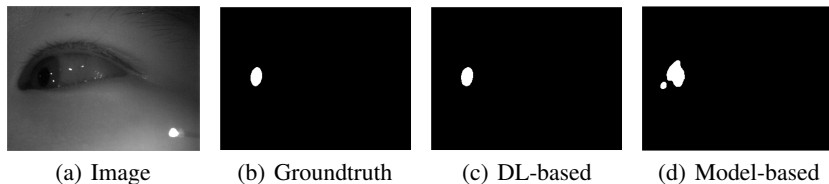

| (a) Image | (b) Groundtruth | (c) DL-based | (d) Model-based |

Figure 11: **An example of eye image with low contrast between pupil and background**. The model-based method fails to extract correct pupil area due to low contrast, while our DL-based method still works correctly compared with the groundtruth mask.

For continuous eye tracking, Figure 12 shows two exemplar trajectory segments obtained by our proposed and model-based method, respectively. The blue dots are pupil centers obtained from the frame-based segmentation module, while the red dots are obtained from the event-based tracking module. We can observe that model-based method produces significantly more problematic "glitches"

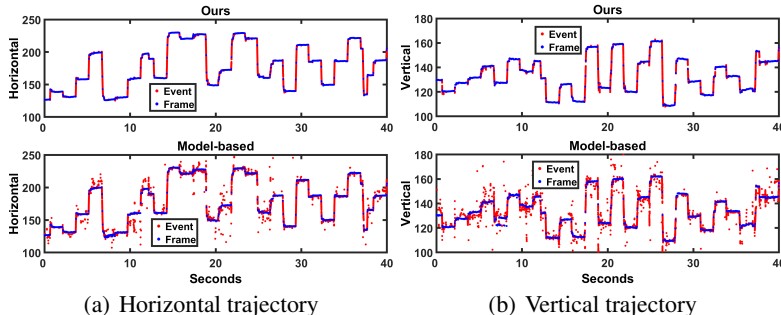

(a) Horizontal trajectory        (b) Vertical trajectory

Figure 12: **Trajectories of pupil tracking under random saccadic state** in horizontal (a) and vertical (b) directions estimated by our method (Ours) and model-based method respectively. The blue dots are from frame-based segmentation and red dots are from event-based tracking.

than our method. According to the discussion in [26], the dataset provided by EVBEYE also contains "glitches", which indicate failed pupil segmentation that causes significant deviation from the correct trajectory. For model-based method, the deviation cannot be corrected until the next correct frame-based segmentation (tens of milliseconds). However, thanks to our new template matching-based method, we can get back on the track as soon as next 20 valid events are accumulated (tens of microseconds).

## D   Study Intrinsic Bias

We discuss some potential intrinsic bias in our datasets. The experiments were recruited within our university, and in total 48 students and staff participated in our study, which admittedly is a relatively small subset of the entire population. We also made sure that no participant from vulnerable groups (such as minors and individuals with health conditions) was involved. Furthermore, with limited resources, currently, we have only recruited participants of Asian/Chinese ethnicity, which may somewhat limit the diversity of the current dataset and introduce population bias. Having that said, to the best of our knowledge, the proposed EV-Eye dataset is currently the largest and most diverse multi-modal frame-event dataset for high frequency eye tracking. We plan to make every effort to expand our dataset in the future, and include more diverse participants. e.g. from different age and ethnic groups.

## E   Intended Uses and Ethical Considerations

The dataset and codes are made solely for research purposes (e.g. replication or further analysis are subject to CC BY-NC 4.0 License and MIT License). Ownership of the original/underlying personal data is vested in the participant himself/herself. Anticipated use cases of this dataset include analyzing how various sensing modalities relate to each other, analyzing how various sensing modalities relate to specific tasks and training learning pipelines that can help extract eye-related features, or analyze eye movement status, etc.

For ethical considerations, consent procedures for data collection were established (for details, please refer to Section G Information Sheet and Consent Form of Participants in the supplementary materials). First, this research does not involve working with vulnerable groups such as minors and people with mental illness. Second, privacy is the major ethical concern of the data collection. One potential risk of multi-modal frame-event datasets is that subjects may be identified using such data without their explicit consent. Therefore, this study strictly follows IRB regulations for anonymizing and keeping participants' data confidential. No one outside the core data collection team has access to the personally identifiable information. In addition, the study ensures participants' autonomy by guaranteeing their right to voluntary participation and withdrawal. Participants engage in the experiment on a voluntary basis and can withdraw at any time during the study without giving a reason. Furthermore, in cases where participants request the removal of their personal data after it is published, any information and data provided by them will be completely destroyed.

# F  Datasheet for Dataset

We adopt the recommended documentation framework Datasheets for Dataset [72] for dataset documentation.

## F.1  Motivation

**For what purpose was the dataset created?** We contribute a first-of-its-kind large scale multi-modal eye tracking dataset to the community in hope to inspire the research on high frequency eye/gaze tracking.

**Who created the dataset?** Team members at AIoT lab, School of Software, Shandong University, China collected, updated and maitain the dataset.

**Who funded the creation of the dataset? If there is an associated grant, please provide the name of the grantor and the grant name and number?** The collection of the dataset is funded by Shandong Provincial Natural Science Foundation, China, Grant No.2022HWYQ-040.

**Any other comments?** [N/A]

## F.2  Composition

**What do the instances that comprise the dataset represent (e.g., documents, photos, people, countries)? Are there multiple types of instances (e.g., movies, users, and ratings;people and interactions between them; nodes and edges)?** The EV-Eye dataset contains multi-modal data collected from two DAVIS346 cameras and one Tobii Pro Glasses 3 when the participants follow the guidance of a stimulus, which are near-eye grayscale images and event streams captured by two sets of DAVIS346 event cameras, and gaze references provided by Tobii Pro Glasses 3.

**How many instances are there in total (of each type, if appropriate)?** The two DAVIS346 cameras produce 1.5 million near-eye grayscale images and more than 2.7 billion events. The Tobii Pro Glasses 3 provides 2.7 million gaze references and 675 thousands scene images.

**Does the dataset contain all possible instances or is it a sample (not necessarily random) of instances from a larger set?** The dataset provided with the link to the reviewer is the full dataset we collected. The full dataset will also be released after acceptance to support the following research.

**What data does each instance consist of? "Raw" data (e.g., unprocessed text or images) or features?** As introduced in Section 3, our dataset provides all the raw data collected by DAVIS346, the gaze reference and scene images output from Tobii Pro Glasses 3. Moreover, we also annotated over 9,000 grayscale images to train and evaluate the pupil segmentation task.

**Is there a label or target associated with each instance? If so, please provide a description** [Yes] The labels are described in **Annotation** part of Section 3.3.

**Is any information missing from individual instances? If so, please provide a description, explaining why this information is missing (e.g., because it was unavailable). This does not include intentionally removed information, but might include, e.g., redacted text.** [N/A]

**Are relationships between individual instances made explicit (e.g., users' movie ratings, social network links)?** [No]

**Are there recommended data splits (e.g., training, development/validation, testing)?** [Yes] . The dataset was collected from multiple sessions across two weeks. The data instances collected from different sessions can be used as training and testing separately. The detailed split of the training and testing can be found from the link to the benchmark for reproducing the results reported in this paper.

**Are there any errors, sources of noise, or redundancies in the dataset?** The event streams contain noisy events caused by lighting of Tobii Glasses and unstable hardware.

**Is the dataset self-contained, or does it link to or otherwise rely on external resources (e.g., websites, tweets, other datasets)?** [Yes] . It is self-contained, all the required dataset and benchmark implementation details can be found from the link provided in Appendix A.

**Does the dataset contain data that might be considered confidential** [No] . The dataset is anonymized and different participants were assigned a number between 1-48 as an ID during data collection.

**Does the dataset contain data that, if viewed directly, might be offensive, insulting, threatening, or might otherwise cause anxiety?** [N/A] .

**Does the dataset relate to people?** [Yes] .

**Does the dataset identify any subpopulations (e.g., by age, gender)?** [N/A] .

**Is it possible to identify individuals (i.e., one or more natural persons), either directly or indirectly (i.e., in combination with other data) from the dataset?** [No] .

**Does the dataset contain data that might be considered sensitive in any way?** [No]

**Any other comments?** [N/A]

### F.3 Collection Process

**How was the data associated with each instance acquired? Was the data directly observable (e.g., raw text, movie ratings), reported by subjects (e.g., survey responses), or indirectly inferred/derived from other data (e.g., part-of-speech tags, model-based guesses for age or language)?** We collected the eye movement of different participants in different moving status including saccade, fixation and smooth pursuit under the guidance of a moving stimulus shown on a display. The detailed procedure of the data collection can be found in Section 3.

**What mechanisms or procedures were used to collect the data (e.g., hardware apparatus or sensor, manual human curation, software program, software API)? How were these mechanisms or procedures validated?** All the procedures and mechanisms are decribed in Section 3.

**If the dataset is a sample from a larger set, what was the sampling strategy (e.g., deterministic, probabilistic with specific sampling probabilities)?** [No] .

**Who was involved in the data collection process (e.g., students, crowdworkers, contractors) and how were they compensated (e.g., how much were crowdworkers paid)?** They are PhD and Master by research students. They were all paid by scholarship from university and salary from our research lab as research assistance.

**Over what timeframe was the data collected?** The dataset was collected from the 1st of July to the 27th of July, 2022 including hardware setup and preliminary test. Most of the dataset published was collected during the last two weeks of July, 2022.

**Were any ethical review processes conducted?** The study was reviewed and approved by the School of Software, Shandong University.

**Did you collect the data from the individuals in question directly, or obtain it via third parties or other sources (e.g., websites)?** The dataset was collected from the individuals in question directly.

**Were the individuals in question notified about the data collection?** [Yes] . The participants signed a consent form before data collection. They were aware of the data collection procedure and the fact the dataset would be made publicly for research purpose.

**Did the individuals in question consent to the collection and use of their data?** [Yes] . All the participants singed a consent form before data collection.

**If consent was obtained, were the consenting individuals provided with a mechanism to revoke their consent in the future or for certain uses?** [N/A] Yes. In the consent form, the participants were informed that they had the right to revoke their consent and withdraw their contribution to the dataset at anytime.

**Has an analysis of the potential impact of the dataset and its use on data subjects (e.g., a data protection impact analysis)been conducted?** [N/A] .

**Any other comments?** [No] .

### F.4 Preprocessing/cleaning/labeling

**Was any preprocessing/cleaning/labeling of the data done (e.g., discretization or bucketing, tokenization, part-of-speech tagging, SIFT feature extraction, removal of instances, processing of missing values)?** [Yes] . The processing and labeling of data was described in Section 3.

**Was the "raw" data saved in addition to the preprocessed/cleaned/labeled data (e.g., to support unanticipated future uses)?** [Yes] . The majority of the dataset consists of raw output from two sets of DAVIS346 event cameras and scene images from Tobii Pro Glasses 3. The dataset also contains processed gaze referencs from Tobii and labeled gray-scale images for benchmarking pupil segmentation.

**Is the software used to preprocess/clean/label the instances available? If so, please provide a link or other access point?** [Yes] . You can find the software and relevant guidance from the website containing the dataset and benchmarks.

**Any other comments?** [N/A] .

### F.5 Uses

**Has the dataset been used for any tasks already?** [No] .

**Is there a repository that links to any or all papers or systems that use the dataset?** [No] . This paper is the first piece of work using this dataset.

**What (other) tasks could the dataset be used for?** Except for high-frequency eye tracking, the dataset could be also used to estimate other dynamic around human eyes, such as wink detection. As the dataset was collected from 48 different participants, the dataset could be also used to study if the eye movement can be used as a robust criterion to differentiate different participants.

**Is there anything about the composition of the dataset or the way it was collected and preprocessed/cleaned/labeled that might impact future uses?** [No] .

**Are there tasks for which the dataset should not be used?** [Yes] . The dataset is for research purpose but cannot be used commercially.

**Any other comments?** [N/A] .

### F.6 Distribution

**Will the dataset be distributed to third parties outside of the entity (e.g., company, institution, organization) on behalf of which the dataset was created?** [No] .

**How will the dataset be distributed (e.g., tarball on website, API, GitHub)?** The source code will be made publicly available on GitHub repository with the download link to the dataset.

**When will the dataset be distributed?** The dataset is distributed at the same time as the paper is published.

**Will the dataset be distributed under a copyright or other intellectual property (IP) license, and/or under applicable terms of use (ToU)?**
The dataset is distributed under a Creative Commons Attribution-NonCommercial 4.0 International License (CC BY-NC 4.0).

**Have any third parties imposed IP-based or other restrictions on the data associated with the instances?** [No] .

**Do any export controls or other regulatory restrictions apply to the dataset or to individual instances?** [No] .

### F.7 Maintenance

**Who is supporting/hosting/maintaining the dataset?** The dataset and source codes of benchmarks are mainted by A-IoT lab, School of Software, Shandong University. The source codes will be hosted on public repository when the paper is published.

**How can the owner/curator/manager of the dataset be contacted (e.g., email address)?** The first author and corresponding author of the paper will be the best persons to be contacted. All the contacts are included in the paper and repository.

**Is there an erratum?** It will be avaible in the GitHub repository.

**Will the dataset be updated (e.g., to correct labeling errors, add new instances, delete instances)?** [Yes] . The dataset will be updated in the repository including new labels and new benchmark methods.

**If the dataset relates to people, are there applicable limits on the retention of the data associated with the instances (e.g., were individuals in question told that their data would be retained for a fixed period of time and then deleted)?** [No] .

**Will older versions of the dataset continue to be supported/hosted/maintained?** [N/A] .

**If others want to extend/augment/build on/contribute to the dataset, is there a mechanism for them to do so?** [Yes] . We will release all our source codes of benchmark and dataset on GitHub Repository so that others can use the dataset to build new methods easily.

**Any other comments?** [N/A] .

### F.8 Author statement

The authors confirm all responsibility in case of violation of rights and confirm the license associated with the dataset (CC BY-NC 4.0 License). The source codes are under MIT License according to https://opensource.org/licenses/MIT.

## G Information Sheet and Consent Form of Participants

In the following pages, we describe the Information Sheet and Consent Form of Participants. Each participant signed the Consent Form before taking part in this study.

School of Software
Artificial Intelligence IoT Laboratory

**Principal Investigator:** Professor Yiran Shen
**Contact:** yiran.shen@sdu.edu.cn
**Primary Researcher:** Guangrong Zhao (DPhil student)
**Telephone number:** +86 17607100755
**E-mail:** guangrong.zhao@sdu.edu.cn

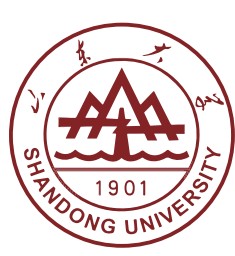

## High Frequency Eye Tracking Based on Event Cameras
### PARTICIPANT INFORMATION SHEET

1. **Background and aims of the study**

   Background: Accurate and high-frequency eye tracking is a key enabler for many game-changing applications, from mental disorder diagnosis, to gaze-based user authentication and virtual reality. However, in practice achieving accurate eye tracking beyond kilohertz frequency is very challenging for the mainstream CCD/CMOS based systems due to the limitations on bandwidth and computation. This has motivated the recent adoption of the emerging bio-inspired event cameras for eye tracking tasks, which can perceive independent pixel-level light intensity changes induced by eye movements at sub-mircosecond latency. While showcasing great potentials over its frame-based counterpart, event-based eye tracking is still in its infancy with many research territories uncharted, and very limited datasets/benchmarks available.

   Aim: The purpose of this research is to explore the potential of event cameras to capture high-frequency eye movement, and to construct a large, diverse multi-modal frame-event dataset for high-frequency eye tracking.

2. **Why have I been invited to participate?**

   The inclusion criteria completely depend on voluntary participation and willingness. Minors and individuals with any known mental health issues are excluded to take part in this trial.

3. **Do I have to participate?**

   You can ask questions about the study before deciding whether or not to participate. If you do agree to participate, you may withdraw yourself and your data from the study at any time, without giving a reason.

4. **What will happen in the study?**

   If you would like to take part in this study, you will be invited to follow the movement of the stimuli on the screen, you will need to wear eyeglasses (eye tracker) and place your head on a chin rest to minimize head movement. Event samples, near eye images and gaze references from wearable eye trackers from you will be recorded.

5. **Are there any potential risks in participating?**

   The gaze information is sensitive data. To prevent leakage of your personal information, we anonymize your names all through the experiments and encrypt all your contact information and collected data. In addition, the data collection procedures are noninvasive and do not pose risk to you. The tasks involved do not involve deception and their completion will not cause you any harm.

6. **Are there any benefits in participating?**

   At the end of the whole experiment, if your data is of good quality, you will be paid 100 RMB.

7. **What happens to the provided data?**

   - Names of the participants are anonymized.

   - Contact information of participants are also encrypted by us and stored on our secure desktops.

8. **Will the research be published?**

   If you agree to participate in this study, the research will be written up as top-tier conference/journal papers at the first stage, then included in the DPhil thesis of the primary research student. It will be published in an open access format and will eventually be available through different channels including the University Library. Information such as the event samples, near eye images and gaze references involved in the study will also be released to the public after removing personal information. You have the right to request the removal of your personal data from the dataset at any time after it is published.

10. **Who do I contact if I have a concern about the study or I wish to complain?**

    If you have a concern about any aspect of this study, please communicate to the relevant researcher Guangrong Zhao, at guangrong.zhao@sdu.edu.cn or his supervisor Yiran Shen ,at yiran.shen@sdu.edu.cn, who will do their best to answer your query.

11. **Contact Details**

    If you would like to discuss the research with someone beforehand (or if you have questions afterwards), please contact:

    Guangrong Zhao

    School of Software

    1500 Shunhua Road, Lixia District, Jinan, Shandong, China 250014

    Tel: +86 17607100755

    Email: guangrong.zhao@sdu.edu.cn

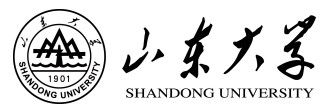

# High Frequency Eye Tracking Based on Event Cameras
# Consent Form for Participants

☐ I am an adult and do not suffer from any known mental health issues.

☐ I have been given a full explanation of this project and have had the opportunity to ask questions.

☐ I understand what is required of me if I agree to take part in the research.

☐ I understand that participation is voluntary and that I may withdraw at any time without giving any reason. Withdrawal from participation also includes the option of withdrawing any information and data I have already provided.

☐ I consent to have event samples, near eye images, and gaze reference data recorded. I understand the format of the data, what information can potentially be extracted from the data, as well as how these data will be stored and publicly released in this study.

☐ I understand that any results published or reported will not reveal my identity information.

☐ I understand that personal information in the data collected for this study will be kept in locked and secure facilities and/or in strong password protected (2FA) electronic form.

☐ I understand that the paper will be published in an open access format and will eventually be available through different channels including the University Library.

☐ I understand the risks associated with taking part and how they will be managed.

☐ I understand that I can contact the researcher Guangrong Zhao guangrong.zhao@sdu.edu.cn or his supervisor Yiran Shen, yiran.shen@sdu.edu.cn for further information.

☐ I consent to my contact information being kept and used by researchers to contact me about future, related research opportunities.

☐ By signing below, I agree to participate in this research project.

Name: _________________ Signed: _________________ Date: _________________

Email address: _______________________________________________