# OpenReview forum: "EV-Eye: Rethinking High-frequency Eye Tracking through the Lenses of Event Cameras"
_NeurIPS.cc/2023/Track/Datasets_and_Benchmarks — NeurIPS 2023 Datasets and Benchmarks Poster_

### Official Review · Reviewer_pW5t · 2023-07-06
**The paper presents a great contribution, which would require some clarification on some parts.**

**Rating:** 8
**Confidence:** 4

**Strengths:**

Promotes further exploration of event-based eye tracking by providing a large and diverse dataset of co-recorded event samples, near-eye images, and gaze references from a wearable eye tracker accompanied by a hybrid algorithm that is easy to understand and implement.

**Additional Feedback:**

Overall, this is a very interesting paper and would help further research on event-based high frequency eye tracking. However, some parts need clarification, mainly what they meant by the “questionability” of the Tobii eye tracker results.

**Clarity:**

The paper was written clearly and concisely without going on tangents or using too complicated words.

**Correctness:**

Dataset and evaluation are comprehensive and thorough.
In the Limitations section (lines 306-307), they point out that the gaze references for Tobii are questionable, particularly for large view angles but never elaborate on it. Clarification on what they mean by this would be helpful.


**Documentation:**

The collection of data has been sufficiently documented. The dataset itself is well-organized and easy to understand. The ReadMe file is particularly useful.


**Ethics:**

I suspect no ethical concerns that warrant further discussion or review. It is also stated that the data collection is approved by their local IRB council.

**Limitations:**

The paper adequately talks about the limitations of the dataset as well as rooms for improvement.

**Opportunities For Improvement:**

Related Work could be more comprehensive as each subsection are very brief.
In line 93, “all three types of eye movements” is confusing as this is not specified until later in section 3.1.
The figures would benefit from more comprehensive captions instead of having the figures be described in the main text, where it is more challenging to find.
In section 5.1, there appear to be no references to Pixel Error and DoD. Are there no previous eye tracking literature that have used these metrics?


**Relation To Prior Work:**

The paper states its difference from previous work by emphasizing on size and robustness of their dataset as well as including more types of eye movements (fixations, saccades and smooth pursuit).

**Summary And Contributions:**

This paper publishes a large and diverse eye tracking dataset with co-recorded samples from a neuromorphic camera and a hybrid eye tracking method that merges eye tracking data with event-based data to significantly boost frequency up to 38.4k Hz.

---

> ### Author Response · Authors · 2023-08-20
> **We appreciate the comments**
>
> **[Related work]**
> We thank the reviewer for the insightful comments. We have provided a more comprehensive discussion on the related work in **Section 2**.
>
> **[Three types of eye movement]**  We agree with the reviewer that the Description of "all three types of eye movements" is unclear. Because we did not define these three types of eye movements beforehand. Therefore, we've changed our description from  “all three types of eye movements”  to  “multiple eye movements i.e., fixation, saccade, and smooth pursuit“ and refer to the place where we define the three types of eye movements.
>
> **[Detailed caption]** We have rewritten the caption of the figures and added more details about the figures.
>
> **[Metrics]** We have provided additional clarification and references in **Section 5.1**.  Pixel Error is a common metric that came from  NVGAZE PupilNetv2.0 and ElSe [1,2,3]. DoD  is a basic indicator similar to the "gaze (angle) estimation error"  on existing eye tracking literature DGTN, ETH-XGaze, and MPIIGaze [4,5,6].
>
> **[Tobii reference]** We have provided a detailed description in **Section 6**.  According to existing studies [7,8], eye-trackers generally are less accurate for large view angles and this seems to be a common issue in many other appearance-based eye-tracking systems such as ETH-XGaze, MPIIGaze, and Angelopoulos et al [5,6,9].
>
> [1] Joohwan Kim, Michael Stengel, Alexander Majercik, Shalini De Mello, David Dunn, Samuli Laine, Morgan McGuire, and David Luebke. Nvgaze: An anatomically-informed dataset for low-latency, near-eye gaze estimation. In Proceedings of the 2019 CHI Conference, New York, NY, USA, 2019. Association for Computing Machinery.
>
>
> [2] Wolfgang Fuhl, Thiago Santini, Gjergji Kasneci, Wolfgang Rosenstiel, and Enkelejda Kasneci. 2017. PupilNet v2.0: Convolutional Neural Networks for CPU based real time Robust Pupil Detection. CoRR abs/1711.00112 (2017). http://arxiv.org/abs/1711.00112
>
> [3] Wolfgang Fuhl, Thiago C Santini, Thomas Kübler, and Enkelejda Kasneci. Else: Ellipse 525 selection for robust pupil detection in real-world environments. In Proceedings of the Ninth 526 Biennial ACM Symposium on Eye Tracking Research & Applications, pages 123–130, 2016.
>
>
> [4]Kang Wang, Hui Su, and Qiang Ji. Neuro-inspired eye tracking with eye movement dynamics. In Proceedings of IEEE/CVF Conference on Computer Vision and Pattern Recognition (CVPR), pages 9823–9832, 2019.
>
> [5] Xucong Zhang, Seonwook Park, Thabo Beeler, Derek Bradley, Siyu Tang, and Otmar Hilliges. ETH-XGaze: A large scale dataset for gaze estimation under extreme head pose and gaze variation. In Proceedings of European Conference on Computer Vision (ECCV), pages 365–381. Springer, 2020.
>
> [6] X. Zhang, Y. Sugano, M. Fritz, and A. Bulling. Mpiigaze: Real-world dataset and deep appearance-based gaze estimation. IEEE Transactions on Pattern Analysis and Machine Intelligence, 41(01):162–175, jan 2019
>
> [7]Accuracy and precision test method for remote eye trackers https://stemedhub.org/resources/3311/download/Tobii_Test_Specifications_Accuracy_and_PrecisionTestMethod_version_2_1_1_.pdf
>
>
> [8] MacInnes, J. J., Iqbal, S., Pearson, J., & Johnson, E. N. (2018). Wearable eye-tracking for research: Automated dynamic gaze mapping and accuracy/precision comparisons across devices. BioRxiv. https://doi.org/10.1101/299925
>
> [9] Anastasios N Angelopoulos, Julien NP Martel, Amit P Kohli, Jörg Conradt, and Gordon Wetzstein. Event-based near-eye gaze tracking beyond 10,000 Hz. IEEE Transactions on Visualization and Computer Graphics, 27(5):2577–2586, 2021.

---

### Official Review · Reviewer_HJTg · 2023-07-15
**A Novel Approach to High-Frequency Eye Tracking Using Bio-Inspired Event Cameras**

**Rating:** 6
**Confidence:** 3

**Strengths:**

- The use of bio-inspired event cameras for high-frequency eye tracking is a novel approach that has the potential to improve  accuracy while effectively tracking the gaze.
- Traditional cameras used in eye tracking are limited by their frame rate, while event-based cameras can adapt the density of events based on the speed of eye movements.
- The EV-Eye dataset created in this study is a valuable resource for researchers interested in gaze tracking.

**Additional Feedback:**

- N/A

**Clarity:**

- The paper is well-organized.

**Correctness:**

- The experiments are designed and conducted in a sound manner.

**Documentation:**

- The dataset is also accompanied by detailed documentation and instructions for use, which enhances its accessibility and usability.

**Ethics:**

- No concerns

**Limitations:**

- The dataset used in this study lacks ground truth for gaze, which makes it difficult to validate the effectiveness of the dataset and evaluate the accuracy of the proposed hybrid eye tracking method. Furthermore, the commercial gaze tracker used in this study were not always reliable.
- Since event cameras are strongly affected by noise, it seems necessary to consider this when acquiring data.
- I doubt whether the dataset is compatible with other equipment or whether the model trained using the data can be used in other equipment

**Opportunities For Improvement:**

In general, the authors have done a good job of addressing the limitations of their work. It would be helpful if the authors mentioned how they would develop their limitations, such as acquiring more accurate gaze references and improving the labeling of eye movement statuses.

**Relation To Prior Work:**

- They provide related works and comprehensive descriptions.

**Summary And Contributions:**

- The paper presents a novel approach to high-frequency eye tracking using bio-inspired event cameras. The authors create a new dataset, EV-Eye, which includes a large number of samples collected from a diverse group of participants. They also develop a hybrid eye tracking method that combines the strengths of event cameras and near-eye grayscale images.

---

> ### Author Response · Authors · 2023-08-20
> **Thank you for your positive feedback.**
>
> **[Groundtruth]** We thank the reviewer for the insightful comments. We have added a description in **Section 3.4** of the revised manuscript for the groundtruth of our dataset: ”In addition to the dynamic gaze references provided by Tobii, EV-Eye also provides the sparse coordinates of PoGs on the monitor screen during fixation states same as those "ground truth" provided by EBVEYE and existing static image-based datasets such as ETH-XGaze [1] and MPIIGaze [2], the sparse coordinates are recorded by scene camera of Tobii Pro Glasses 3 eye tracker synchronously.”  Therefore, our data set not only contains references from eye trackers but also contains "ground truth" similar to traditional static image-based datasets. The main purpose of using an eye tracker is to provide more detailed phases of eye movement , and the inspiration for building the dataset using the tobii eye tracker came from some existing Video-based Eye-Tracking work such as DGTN dataset and  EVE dataset [3][4].
>
> [1] Xucong Zhang, Seonwook Park, Thabo Beeler, Derek Bradley, Siyu Tang, and Otmar Hilliges. ETH-XGaze: A large scale dataset for gaze estimation under extreme head pose and gaze variation. ECCV, pages 365–381. Springer, 2020.
>
> [2] X. Zhang, Y. Sugano, M. Fritz, and A. Bulling. Mpiigaze: Real-world dataset and deep appearance-based gaze estimation.TPAMI, 41(01):162–175, jan 2019
>
> [3] Kang Wang, Hui Su, and Qiang Ji. Neuro-inspired eye tracking with eye movement dynamics. CVPR, 473 pages 9823–9832, 2019.
>
> [4] Seonwook Park, Emre Aksan, Xucong Zhang, and Otmar Hilliges. 2020. Towards End-to-End Video-Based Eye-Tracking. ECCV, Berlin, Heidelberg, 747–763.
>
>
> **[Noise filtering]** We agree with the reviewer that in general, event data is more noisy than traditional frame data. We do not denoise events in the preprocessing stage. Instead, we constructed the Candidate Points Subset in **Section 4.2**, which enables us to filter out noisy events which are unrelated to the actual pupil movement.
>
> **[Compatible with other equipment]** We appreciate the comments. We have not yet tested our model on various devices. However, when we tested our model on EBVEYE (a dataset collected by another type of event camera DAVIS346B), we found that only a small amount of data (less than 100 samples) was needed to fine-tune the model to achieve an IOU index of more than 90 for segmenting pupil area.

---

> > ### Comment · Reviewer_HJTg · 2023-08-27
> >
> > Thank you for your effort in addressing my concerns. I grateful for the inclusion of ground truth data to enhance the precision of the eye tracking, as well as for the thorough verification of compatibility with other equipment.
> >
> > I expect EV-Eye datset will contribute to its utility in the field of eye tracking, since there aren't many event camera-based datasets.
> >
> > My concerns have been appropriately addressed. I intend to adjust my rating while observing the responses from the other reviewers.

---

### Official Review · Reviewer_MbfV · 2023-07-21
**Event based eye tracking dataset along with consumer grade Tobii tracker as a reference**

**Rating:** 6
**Confidence:** 3
**Correctness:** Yes, the paper's claims seems to be r…

**Strengths:**

1. Doubles the number of participants from 24 to 48 subjects.
2. Demonstrates tracking capabilities that nearly triples the earlier claims from around 10kHz to 3.8 kHz

**Additional Feedback:**

Demonstrating the improvement in task level performance due to the enhancement in sampling frequency could strengthen this paper significantly.

**Clarity:**

Yes, the paper is well written although some more details and better contrasting with earlier studies could help.

**Documentation:**

The paper and dataset are reasonably documented.

**Ethics:**

No concerns expect in cases where pupil images are considered personally identifiable information.

**Limitations:**

1. Does not collect ground truth explicitly and relies on a commercial grade tracker as reference, this would incorporate the errors made by the Tobii tracker into the reference.
2. Does not explicitly provide any use cases or instances where increased sampling frequency has unblocked any applications
3. Use of the dataset has not been demonstrated for any tasks
4. Does not provide comprehensive analysis on factors that have contributed to improved the effective sampling rate compared to earlier attempts by Angelopoulis et. al.

**Opportunities For Improvement:**

1. The acquisition protocol is very similar to "Event-Based Near-Eye Gaze Tracking Beyond 10,000 Hz" and paper does not dive deep into the differences that have lead to improved data quality compared to this study.
2. Lines 152-163 could be elaborated to deep dive into the factors that contributed to improvements in the performance, for instance while using the same/similar event sensors how the current study almost tripled the effective sampling rate.
3. Caption for Figure 2 could provide more details.

**Relation To Prior Work:**

Yes, the paper cites relevant work.

**Summary And Contributions:**

The papers presents an eye tracking dataset using event cameras instead of the traditional frame based sensors. A protocol very similar to an earlier study is adapted and fixation and smooth pursuit moments are collected from 48 participants. Multi modal data (image frames at 25Hz and event streams) from two Davis346 cameras as well as reference eye tracking data from Tobii Glasses 3 are provided. This study  has doubled the number of subjects from which similar data is available and also pushes the dynamic sampling frequency up to 38.4 kHz from 10kHz reported in earlier studies. The paper also claims to have improved the Gaze prediction accuracy although reference using Tobii is not very robust.

---

> ### Author Response · Authors · 2023-08-20
> **We thank the reviewer for the insightful comments.**
>
> **[Gaze reference]** We agree with the reviewer that the results from Tobii are references rather than groundtruth. According to our observation,  tobii eye tracker has questionable results at large viewing angles like existing studies in ETH-XGaze[1] and MPIIGaze [2].  However, in most cases,  it can achieve less than 0.6 angular error in gaze estimation. There is other work utilizing tobii eye tracker as reference to construct gaze tracking dataset [3][4].
>
> [1] Xucong Zhang et al. ETH-XGaze: A large scale dataset for gaze estimation under extreme head pose and gaze variation. ECCV, pages 365–381. Springer, 2020.
>
> [2] X. Zhang, et al. Mpiigaze: Real-world dataset and deep appearance-based gaze estimation. TPAMI, 41(01):162–175, jan 2019
>
> [3] Kang Wang, et al. Neuro-inspired eye tracking with eye movement dynamics. CVPR, pages 9823–9832, 2019.
>
> [4] Seonwook Park, et al. Towards End-to-End Video-Based Eye-Tracking. ECCV 2020, Berlin, 747–763.
>
> **[Difference to EVBEYE]** As suggested, we have highlighted the differences between our dataset and EBVEYE in **Subsection 3.4**. The main difference between our dataset and the EBVEYE dataset is that our dataset is larger in size and contains more eye movement state references and annotations of pupil regions.
>
> **[Improved sampling rate]** The effective sampling rate depends on how fast the eye is moving and the refreshing condition (i.e., how many events need to be accumulated). Our update frequency is calculated as follows: the time difference between the first and last event in the Candidate Points Subset is T and the instantaneous tracking frequency is defined as F = 1/T. It is different from the one used in EVBEYE, EVBEYE calculated the update frequency by counting the number of events around the eye per unit time. If the same calculation method is used, the effective sampling rate of EBVEYE can also be up to 38.4KHz when processing our dataset.
>
> **[Caption of Figure 2]** As suggested, we have added a more detailed description in the **caption of Figure 2** : “Compared to the EBVEYE dataset， the gaze references provided in our dataset are significantly more dense and involve all states, i.e., fixation, saccade and smooth pursuit.”
>
> **[Groundtruth]** We thank the reviewer for the insightful comments. Our description of ground truth was not clear enough. We have described the groundtruth we collected in **Section 3.4 of the revised manuscript**: ”In addition to the dynamic gaze references provided by Tobii, EV-Eye also provides the sparse coordinates of PoGs on the  monitor screen during fixation states same as those "ground truth" provided by EBVEYE and existing static image-based datasets such as ETH-XGaze and MPIIGaze, the sparse coordinates are recorded by scene camera of Tobii Pro Glasses 3 eye tracker synchronously.”  Therefore, our data set not only contains references from eye trackers but also contains similar groundtruth to traditional static image-based datasets [1] [2]. The main purpose of using an eye tracker is to provide more detailed phases of eye movement as it can keep track of the motion. Moreover,  in most cases,  tobii eye tracker can achieve less than 0.6 angular error in gaze estimation[3], and there is other work utilizing tobii eye trackers to construct Video-based Eye-Tracking dataset [4][5].
>
> **[Applications and Tasks Support]**  We agree with the reviewer that including applications in the dataset will be a great improvement. However, as discussed in the paper, our current focus on this dataset is to inspire future research on designing new algorithms or systems for high frequency eye tracking with event camera.  The multi-modal frame-event dataset can be used to develop various eye-tracking systems based on event cameras, and then serve various applications such as mental disorders diagnosis and VR. High frequency eye tracking is great for mental health diagnosis. However,  at the current stage, for ethical considerations, our dataset does not include vulnerable groups such as people with Alzheimer's disease, which is common in the elderly population, and ADHD, which is common in the pediatric population.
>
> **[Higher frequency]** Our previous description was not clear enough. As mentioned above, the effective sampling rate depends on how fast the eye is moving and the refreshing condition (i.e., how many events need to be accumulated). The high frequency is because of the high temporal resolution of the event camera and we did not claim it as the contribution of our algorithm. In fact, if we use the eye tracking method of EVBEYE proposed in Angelopoulis et. al. It can also achieve the same high frequency as ours in our dataset.
>
> **[Contrasting with earlier studies]** As suggested, we have added a new subsection, **”Dynamic Vision Sensors”**  in **Section 3**  to make the paper self-contained and we have added a description that outlines the difference between our dataset and earlier EBVEYE dataset **at the end of Section 3.4.**

---

> > ### Comment · Reviewer_MbfV · 2023-08-26
> > **Setting expectations in the abstract**
> >
> > Thank you for your response. I have only one minor comment in the context of the above responses. Reading the abstract with a sentence like "Our method can track pupil locations at a frequency of up to 38.4kHz, outperforming the state-of-the-art baseline significantly." I have expected significant differences in the tracking method compared to the literature. It made me think there were advances to the experimental protocol or use of novel sensors etc. It would be great if the abstract can closely reflect on the additional information provided by this dataset.

---

> > > ### Author Response · Authors · 2023-08-28
> > >
> > > Thanks for the comment, the reviewer is right, this sentence is indeed misleading, the tracking frequency of 38.4khz is not a contribution of our algorithm but a benefit from the event camera, we will rephrase this sentence in the revised paper.

---

> > > > ### Comment · Reviewer_MbfV · 2023-08-29
> > > > **Rating remains the same.**
> > > >
> > > > I hope the authors would revise their abstract, the recent version is similar to the earlier version with claims about the methods ability to track at higher frequencies. My major concerns on the utility of the signal at very high frequencies (do we have good signal/noise ratio in that regime?) and novelty remain. I believe the protocol and sensor used are close to the earlier methods. However, as this dataset increases the volume of data available with a similar protocol it is still very useful.

---

> > > > > ### Author Response · Authors · 2023-08-30
> > > > > **About the revision of the abstract**
> > > > >
> > > > > Thanks for the suggestion, we misunderstood the reviewer before, so we only deleted the sentence describing the high-frequency algorithm that was misleading. **We have revised the abstract** by adding the following description: "Compared with existing event-based high-frequency eye tracking datasets, our dataset is significantly larger in size, and **the gaze references involve more natural eye movement patterns**, i.e., fixation, saccade and smooth pursuit." **This highlights the improvement of our dataset compared to earlier dataset** i.e., EVBEYE.
> > > > >
> > > > > **For the algorithm contribution**, we provide two benchmark methods for comparison, i.e., our DL&Matching-based method  and the SOTA Model-based method provided by Angelopoulos, et al.  From the evaluation results, our method is more robust and has higher tracking accuracy. **Thus, the contribution of our algorithm is not to improve the frequency of eye tracking but to improve the tracking accuracy and robustness.** **We have also added the following description in the abstract**: "We show that our method achieves higher accuracy for both pupil and gaze estimation tasks compared to the existing solution."

---

> > > > > > ### Comment · Reviewer_MbfV · 2023-08-30
> > > > > > **No further comments**
> > > > > >
> > > > > > Thank you for making the changes. Looking forward to the application level advancements this data set could unlock.

---

### Official Review · Reviewer_MqwV · 2023-07-21
**In this paper, the authors presented EV-Eye dataset, a diverse event-based multimodal dataset for high-frequency eye tracking, collected from 48 subjects with different devices. It proposes a novel hybrid frame-event eye tracking approach achieving a tracking frequency of up to 38.4Hz and demonstrates superior accuracy and robustness compared to the state-of-the-art method through extensive evaluations of EV-Eye. It can potentially fit into the Track Datasets and Benchmarks. This dataset could provide researchers to implement multiple real-time applications and deep learning methods.**

**Rating:** 6
**Confidence:** 4
**Correctness:** Claims made in the submission seem sa…

**Strengths:**

This work has a significant contribution and can help the broader research community.

**Additional Feedback:**

More details on accurate annotation and ground truth would be helpful for broader community use. Authors should add the labelling gaze locations, eye movement types, and other relevant information for training and evaluation purposes.
Authors should include participant demography inclusion of vision correction, age range etc.

**Clarity:**

There are grammatical mistakes, and some phrases must be more concise throughout the manuscript.

**Documentation:**

The dataset organization and folder structure, which include processed data, raw data, code scripts, and a comprehensive readme file, have been well-organized and presented satisfactorily.

**Ethics:**

The authors should add information about the explicit consent of participants.
Authors must respect the terms of datasets that have defined licenses (e.g. CC 4.0, MIT, etc). Somehow this is missing.

**Limitations:**

The authors mention the limitations of the work in the paper.

**Opportunities For Improvement:**

The dataset should consider the inclusion of a diverse range of subjects in the dataset to account for individual variations in eye movements, including different age groups, genders, and ethnicities. This ensures the dataset's generalizability.
Another point – it would be interesting if the dataset considers various real-world scenarios to make the dataset representative of natural eye-gaze behaviours. This may include activities like reading, watching videos, driving, or interacting with objects.


**Relation To Prior Work:**

Relation to prior work is explained well, and comparing other work is defined.

**Summary And Contributions:**

This work presents a diverse event-based multimodal dataset for high-frequency eye tracking and a hybrid frame-event eye tracking benchmarking approach tailored to the collected dataset, capable of tracking the pupil at a frequency up to 38.4kHz.

---

> ### Author Response · Authors · 2023-08-20
> **Thank you for your positive feedback.**
>
> **[Subject Diversity]** We agree with the reviewer that including subjects from diverse age groups, genders, and ethnicities can certainly improve the generalizability of our dataset. Admittedly, our current dataset was collected in a university context, with students/staff (28 male and 20 female) from age groups between 21 and 35, of primarily Asian ethnicity, which may be slightly biased in some sense (e.g. no very diverse ethnicities). We plan to make every effort to expand our dataset in the future and include more diverse participants.  We have clarified this in the newly added **Study Intrinsic Bias in Section D of the supplementary material.**
>
> **[Real-world scenarios]** It is a valuable suggestion to improve the quality and generalization of our dataset in the future, considering various scenarios in the real world can enrich the gaze behavior included in the dataset. In the context of the current work, we selected three **major types** in natural viewing scenarios, i.e., fixation, saccade and smooth pursuit [1,2,3]. They encompass the vast majority of human eye movements and can be used for promoting the further design of event-based eye tracking algorithms. We will keep improving our dataset, in the next stage, we plan to use video stimuli such as CRCNS, DIEM, and Gaze360 [4] in the wild environment to capture more diverse types of eye movements.
>
> [1]https://www.ncbi.nlm.nih.gov/books/NBK10991/
>
> [2] Komogortsev, O.V., Karpov, A. Automated classification and scoring of smooth pursuit eye movements in the presence of fixations and saccades. Behav Res 45, 203–215 (2013). https://doi.org/10.3758/s13428-012-0234-9
>
> [3]Kang Wang, Hui Su, and Qiang Ji. Neuro-inspired eye tracking with eye movement dynamics. 472 In Proceedings of IEEE/CVF Conference on Computer Vision and Pattern Recognition (CVPR), 473 pages 9823–9832, 2019
>
> [4]Petr Kellnhofer*, Adrià Recasens*, Simon Stent, Wojciech Matusik, and Antonio Torralba. “Gaze360: Physically Unconstrained Gaze Estimation in the Wild”. IEEE International Conference on Computer Vision (ICCV), 2019.
>
> **[Grammar mistakes]** We will carefully revise our paper and iron out any grammar issues throughout.
>
> **[License]** We have added detailed information about the  **Information sheet** and **consent form**  in  **supplementary material**,  it describes the informed consent process for participants, and includes descriptions of data use and voluntary participation and withdrawal.  The author statement about defined licenses has also been added in **Section F.8 of the supplementary material**: “The authors confirm all responsibility in case of violation of rights and confirm the licence associated with the dataset (CC BY-NC 4.0 License). The source codes are under MIT License according to https://opensource.org/licenses/MIT.”
>
>
> **[More labels]** We agree with the reviewer that adding the labeling gaze locations, eye movement types, and other relevant information for training and evaluation purposes will broaden community use. We have added 19270 new labels with pupil, iris, and eye movement types including saccade, smooth pursuit, and blinks to the folder EV_Eye_dataset/raw_data/Data_davis_pupil_iris_label **in our dataset link: https://1drv.ms/f/s!Ar4TcaawWPssqmu-0vJ45vYR3OHw.** In addition, we have added information about participants’ vision correction, age range into the **second paragraph of Section 3.3， and this information for each individual is also uploaded into our dataset link.**

---

> > ### Comment · Reviewer_MqwV · 2023-08-30
> >
> > I appreciate the authors for addressing most of the comments. I recommend that the authors consider adding a limitation section to the paper, discussing aspects such as constraints related to real-world scenarios and other overall limitations.
> >
> > I suggest authors to see NeurIPS 2023 guidelines/code of conduct. Authors are not permitted to compel reviewers to respond during the rebuttal period.

---

> > > ### Author Response · Authors · 2023-08-31
> > > **About adding more discussion to the limitation section**
> > >
> > > Thanks for the suggestion. We've added more discussion of subject diversity and real-world scenario limitations in Section 6 of the revised paper.

---

### Official Review · Reviewer_QL5M · 2023-07-27
**EV-Eye: Rethinking High-frequency Eye Tracking through the Lenses of Event Cameras**

**Rating:** 7
**Confidence:** 2
**Correctness:** See above.
**Clarity:** The paper is generally clear.

**Strengths:**

The multimodal nature of the dataset and the claimed temporal accuracy are both significant strengths.


**Additional Feedback:**

Please discuss the nature of the event camera data in more detail.

**Documentation:**

As far as I was able to tell, yes.

**Ethics:**

No issues.

**Limitations:**

See above.

**Opportunities For Improvement:**

A more detailed explanation of event cameras and the claimed 1/38,400 second precision would improve the paper. It's not clear to me that the claim of "tracking the pupil at a frequency up to 38.4kHz" is correct because, as I understand it, the data is not actually sampled 38,400 times a second.

I also note that the authors state "However, the high frame-rate presents a considerable computational burden on downstream processing, rendering it impossible for VR headsets." as a limitation of 1000 Hz data, which seems a contrived problem.

**Relation To Prior Work:**

Yes.

**Summary And Contributions:**

The paper describes a new dataset of multimodal eye tracking data from 48 subjects.

---

> ### Author Response · Authors · 2023-08-20
> **Thank you for your positive feedback.**
>
> **[Tracking Frequency]** The tracking frequency in our system is not at a fixed rate: it is determined by how long it takes to accumulate 20 events caused by the movement of the pupil region. Therefore, the tracking frequency is **adaptive** and proportional to the movement speed.  38.4KHz is the maximum tracking frequency calculated in our dataset (in saccade state). This is not because of our algorithm, the model-based method in EVBEYE can also achieve 38.4KHz in our system.  In fact, the extremely-high tracking frequency is an inherent property of an event camera which is high in temporal resolution (around **tens of microseconds**).  **We have added the above description in the second paragraph of Section 5.4.**
>
> **[Computational burden]** We agree with the reviewer that our statements on computational burden may not be appropriate. We have changed it to: "However, the high frame-rate presents a considerable computational burden on downstream processing, rendering it **unsuitable** for VR headsets." Here we want to emphasize the much more excessive computation when using traditional high-speed cameras than event cameras, as the former needs to process every frame to capture eye movements, while the latter only outputs sparse events and our algorithm can be fully adaptive with respect to the speed of eye movements. Therefore in practice our approach typically doesn’t need to constantly work at a very high frequency: when eyes are in fixed or slow-moving states, event cameras will only generate relatively few events, leading to much fewer updates, and thus better utilization of compute, energy and bandwidth.
>
> We've added the description of the **adaptive update strategy in the revised version** to illustrate that the event-based pupil tracking system can better utilize the resources in  **Section 5.4.**
>
> **[Nature of event camera]** To illustrate the nature of event camera data,  we have added the **description of event camera data in  Section 3.1.**

---

### Author Response · Authors · 2023-08-20
**We would like to thank the editors and reviewers for their insightful comments and suggestions, and the diligence in reviewing our work. In response,we have thoroughly modified our manuscript as requested and addressed the key concerns raised by the reviewers.**

Regarding **ethical concerns**, we have provided additional information required by the reviewers in our response below (as well as responses to individual comments) and clarified these points in the revised paper. The **Information Sheet** and **Consent Form** have been incorporated into **Section G of the  Supplementary Material of the revised manuscript.**  In addition, the **Intended Uses and Ethical Considerations** have been added to **Section E of the Supplementary Material of the revised manuscript**, while the **Study Intrinsic Bias**  has also been discussed in **Section D in  Supplementary Material of the revised manuscript.** The changes and additional information are summarized as follows:

In the data collection process, participants voluntarily participated in our study and could withdraw at any time without any reason. They were aware that the dataset and the paper would be made publicly available after removing personally identifiable information. In addition, participants also had the right to request the removal of their personal data from the dataset at any time after it is published. The final ownership of the data belonged to the participants themselves, and our dataset and code are licensed under the CC BY-NC 4.0 License and MIT License, intended solely for research purposes. Additionally, the Study Intrinsic Bias discussed the biases in the dataset. The study was conducted within our university, and in participants, recruitment stage volunteers from vulnerable groups (e.g. those identified themselves with disabilities or invisible disabilities) were excluded. Regarding data diversity, we are aware that at the moment our data has been collected only from an academic institution (potentially with population bias): we will try our best to expand our dataset to include    participants from different age groups and ethnic backgrounds in the future. Having that said, to the best of our knowledge, the proposed EV-Eye dataset is currently the largest and most diverse multi-modal frame-event dataset for high frequency eye tracking.

Regarding other comments, we have tried to address those in our responses and elaborate the improvements made in the revised manuscript. We have added a description of the **nature of event camera**, added explanations about **tracking frequency**, discussed the **subject diversity** of the dataset, provided a description of the **gaze ground truth** contained in our dataset, added **contrasting with earlier studies**, and delved into the discussion of **applications and task support** of our dataset.  Please kindly refer to our responses to individual comments below for more details.

---

### Decision · Program_Chairs · 2023-09-22

**Decision:**

Accept (Poster)

**Comment:**

The paper presents a new multimodal dataset of eye movements captured with an event-based camera and a grey-scale frame camera. The dataset excels in featuring sub microsecond latency. Also, in comparison with the existing datasets it is larger in size and the gaze references include more natural eye movement patterns (fixation, saccade, smooth pursuit). The paper also presents a new hybrid method for eye tracking, leveraging the strengths of the event-based cameras and near-eye grayscale images, as a benchmark. The initial assessments of the quality of the paper by all five reviewers have mostly been positive, stressing the importance and size of the dataset. While clarity and correctness have predominantly been judged positively, several issues have been raised, which required clarifications and/or additions and revisions. In response, the authors provided the requested clarifications (e.g., related to the ground truth, reference systems) and added additional labels. In addition to the issues raised with respect to the technical content, some of the reviewers also questioned whether all the ethical considerations had been taken into account and the process had followed the established ethical procedures for this type of data. The authors have provided further clarifications and also included the relevant documents in the Supplementary material. At the end of the rebuttal process, the reviewers assessed that their concerns (technical and ethical) have adequately been addressed and they recommended acceptance of the paper.